# Resolution dependency of sinking Lagrangian particles in ocean general circulation models

**Peter D. Nooteboom**[1,2]*, **Philippe Delandmeter**[1], **Erik van Sebille**[1,2], **Peter K. Bijl**[3], **Henk A. Dijkstra**[1,2], **Anna S. von der Heydt**[1,2]

**1** Department of Physics, Institute for Marine and Atmospheric Research Utrecht (IMAU), Utrecht University, Utrecht, Netherlands, **2** Centre for Complex Systems Studies, Utrecht University, Utrecht, Netherlands, **3** Laboratory of Paleobotany and Palynology, Department of Earth Sciences, Utrecht University, Utrecht, Netherlands

* p.d.nooteboom@uu.nl

**Data Availability Statement:** All the code that can be used to reproduce the figures/results in the manuscript can be found on github: https://github.

## Abstract

Any type of non-buoyant material in the ocean is transported horizontally by currents during its sinking journey. This lateral transport can be far from negligible for small sinking velocities. To estimate its magnitude and direction, the material is often modelled as a set of Lagrangian particles advected by current velocities that are obtained from Ocean General Circulation Models (OGCMs). State-of-the-art OGCMs are strongly eddying, similar to the real ocean, providing results with a spatial resolution on the order of 10 km on a daily frequency. While the importance of eddies in OGCMs is well-appreciated in the physical oceanographic community, other marine research communities may not. Further, many long term climate modelling simulations (e.g. in paleoclimate) rely on lower spatial resolution models that do not capture mesoscale features. To demonstrate how much the absence of mesoscale features in low-resolution models influences the Lagrangian particle transport, we simulate the transport of sinking Lagrangian particles using low- and high-resolution global OGCMs, and assess the lateral transport differences resulting from the difference in spatial and temporal model resolution. We find major differences between the transport in the non-eddying OGCM and in the eddying OGCM. Addition of stochastic noise to the particle trajectories in the non-eddying OGCM parameterises the effect of eddies well in some cases (e.g. in the North Pacific gyre). The effect of a coarser temporal resolution (once every 5 days versus monthly) is smaller compared to a coarser spatial resolution (0.1˚ versus 1˚ horizontally). We recommend to use sinking Lagrangian particles, representing e.g. marine snow, microplankton or sinking plastic, only with velocity fields from eddying Eulerian OGCMs, requiring high-resolution models in e.g. paleoceanographic studies. To increase the accessibility of our particle trace simulations, we launch planktondrift.science.uu.nl, an online tool to reconstruct the surface origin of sedimentary particles in a specific location.

## Introduction

Sinking particles are involved in fundamental processes in the ocean. They serve as a primary mode of carbon export out of the exogenic carbon pool and deliver sediment to the world

com/pdnooteboom/PO_res_error. The results of this paper can also be accessed on planktondrift. science.uu.nl.

**Funding:** This work was funded by the Netherlands Organization for Scientific Research (NWO), Earth and Life Sciences, through project ALWOP.207. The use of the SURFsara computing facilities was sponsored by NWO-EW (Netherlands Organisation for Scientific Research, Exact Sciences) under the project 17189. The European Research Council under the European Community's Seventh Framework Program provided funding for this work by ERC Starting Grant #802835 (OceaNice) to PKB. PD and EvS are supported through funding from the European Research Council (ERC) under the European Union Horizon 2020 research and innovation programme (grant agreement no. 715386, TOPIOS).

**Competing interests:** The authors have declared that no competing interests exist.

ocean floor: An important archive for understanding the climate system. The lateral advection of the sinking particles by ocean currents complicates the estimation of downward particle fluxes captured by sediment traps [1], the paleoceanographic reconstructions based on sedimentary microplankton distributions [2–5], and the estimation of micro-plastic distributions in the ocean [6]. Initially buoyant micro-plastic in the ocean sinks when it gets biofouled and its density increases [7], meaning that a large fraction of the plastic in the ocean has already sunk to the ocean floor [8]. The lateral transport of sinking particles can be estimated using Ocean General Circulation Models (OGCMs) and Lagrangian tracking techniques [9]. The Lagrangian techniques are used to model the sinking particle trajectories in the modern ocean [10–15], specifically for sinking microplankton [16, 17] and microplastic [18].

Where possible, these Lagrangian techniques make use of an eddying flow field. However, eddying simulations are not available for all applications to provide such a flow field. For example, model simulations of the geological past use OGCMs with at most 1° horizontal (non-eddying) resolution [19–23]. The latter is due to the fact that palaeoclimate model simulations require coupled climate model simulations (because the atmospheric forcing is not known from observations) and long spin-up times (typically a few 1000 model years) in order to reach a reasonable climate equilibrium.

The spatial and temporal resolution of the underlying flow field generated by OGCMs will affect the spreading of particles in the Lagrangian tracking. It has already been shown that Lagrangian trajectories of neutrally buoyant particles are sensitive to the temporal resolution in an OGCM with $\sim 2°$ horizontal resolution [24], and the temporal resolution influences the divergence timescale of trajectories in an OGCM of 0.1° horizontal resolution [25].

The spatial resolution of the OGCM determines if the flow is eddying, which played an important role in simulations of sinking particles near the northern Gulf of Mexico [26] and in the Benguela region [13], and for passive tracers near Sellafield [27] and globally [28] (0.25° versus 1° resolution). Eddying OGCMs generate a different time-mean flow compared to non-eddying OGCMs which parameterise the eddy effects [29, 30]. The interplay between eddies and the mean flow is found to be important for the representation of internal variability of the flow (i.e. the variability of the system under constant atmospheric forcing) [31]. This results in a better representation of interannual or multidecadal variability [32] and the separation location of western boundary currents such as the Gulf Stream [33]. Additionally, eddies cause mixing of tracers (e.g. heat and salinity). The non-eddying OGCMs rely on parameterisations of this tracer mixing such as the Gent-McWilliams (GM) parameterisation [34, 35], which shows difficulties to represent this effect locally [36, 37].

In this paper, we will assess how the sinking Lagrangian particle trajectories vary for different temporal or spatial resolutions of an Eulerian OGCM. We investigate the effect of eddies on the particle trajectories. Moreover, we study whether a stochastic lateral diffusion of Smagorinsky [38] type could parameterise the effects of the eddies in the non-eddying OGCM. We use the same analysis as is applied in [17] about microplankton which is used for palaeoceanographic reconstruction (specifically dinoflagellate cysts). The results concern any type of application with sinking Lagrangian particles, such as the comparison of sediment trap data with OGCMs [15, 26] or the representation of sinking microplankton [16, 17] and sinking plastic [7].

We disseminate our results further with an interactive website: https://www.planktondrift. science.uu.nl. This online tool simulates the surface origin of particles that sink to the bottom of the present-day ocean. The tool can also be used to determine how the microplankton in the bottom sediments at any location of choice relates to the environment at these origin locations (e.g. temperature, salinity, primary productivity) in the present-day ocean (see also [17]).

## Method

We make use of present-day global ocean model simulations of the Parallel Ocean Program (POP) with 0.1° ($R_{0.1}$; eddying) and 1° ($R_{1m}$; non-eddying) horizontal resolution to advect virtual particles (also used in [37, 39, 40]). The eddying POP version has a reasonably good representation of the modern circulation compared to other models at the same resolution [41]. Both versions of POP are configured to be as consistent as possible with each other, but there are some differences (see the supplementary material of [39]).

   We apply the same particle tracking approach as in [17]. This means that we release particles at the bottom of the ocean every three days for more than a year, and compute their trajectories in the changing flow field back in time (similar to [9, 15, 17, 26]) until the particles reached the surface. We stop a particle if it reaches 10m depth. The particles are released on a 1° × 1° global grid. The resulting particle distributions allow us to investigate the statistics of particle ensembles, rather than single trajectories. Particle ensemble statsistics are often used in Lagrangian analysis [9], because of the chaotic nature of the particle trajectories [42]. While the particles are advected back in time, a constant sinking velocity $w_f$ is added to the particle trajectories. The addition of a constant sinking velocity to an advected particle has been shown to be a proper way to incorporate the effect of gravity on a sinking particle [43]. We used Parcels version 2.0.0 [44] to calculate the particle trajectories, which is compatible with the Arakawa B-grid that POP uses.

   The sinking velocity of particles in the ocean varies substantially. The sinking speed $w_f$ of microplastics is on the order of 3.4-50 m day$^{-1}$ [8], for single dinoflaggellate cysts $w_f$ ranges from 6–11 m day$^{-1}$ [45], and the sinking speed can become several hundreds of meters per day for marine snow aggregates (e.g. 10-287 m day$^{-1}$ [46]). The larger $w_f$, the shorter the travel time of the particles will be and the less the particle distributions at the surface will spread. Here, we focus on two sinking speeds: $w_f$ = 6 and 25 m day$^{-1}$, to study the dependence of the results on the sinking speed, i.e. we represent the sinking of individual dinoflagellate cysts and small aggregates, respectively. More scenarios of sinking speed $w_f$ were investigated in [17].

   The particle trajectory is integrated using the velocity field of POP and a stochastic term parameterising the effect of unresolved processes on the velocity. This last term is equivalent to diffusion in Eulerian models [9] and is a function of the diffusivity $v$. Here we define $v$ as a function of the mesh size (i.e. the size of the grid cell) and the flow shear, following the Smagorinsky [38] parameterisation, which is commonly used in OGCMs and Large Eddy Simulations (LES). This implies that the particle trajectories are computed by:

$$\vec{x}(t - \Delta t) = \vec{x}(t) + \int_{t}^{t-\Delta t} \vec{v}(\vec{x}, \tau)d\tau + \vec{c}\Delta t + \vec{q}\sqrt{2v(\vec{x})\Delta t}, \qquad (1)$$

with $\vec{x}(t)$ the three-dimensional position of the particle at time $t$, $\vec{v}(\vec{x}, t)$ the flow velocity at location $\vec{x}$ and time $t$ (linearly interpolated in space and time from the flow field), and $\vec{c} = \begin{pmatrix} 0 & 0 & -w_f \end{pmatrix}^T$ the sinking velocity. The vertical part of the flow $\vec{v}$ can be relevant compared to the particle sinking velocity $w_f$ (see Fig 7 in [17]). The flow consists of two components in the non-eddying POP model: $\vec{v} = \vec{v}_a + \vec{v}_b$, where $\vec{v}_a$ is the Eulerian flow field that is solved by POP. $\vec{v}_b$ is the bolus velocity from the GM parameterisation, which represents the flow that is responsible for the mixing of tracers along isopycnals [34, 35]. $\vec{v}_b = \vec{0}$ in the eddying POP model.

   The last term of Eq 1 is the horizontal diffusivity term (only used in the non-eddying model), where $\vec{q} = \begin{pmatrix} R_1 & R_2 & 0 \end{pmatrix}^T$ represents (independent) white noise in the zonal and

meridional direction, with mean $\mu_{R_1} = \mu_{R_2} = 0$ and variance $\sigma^2_{R_1} = \sigma^2_{R_2} = 1$, and

$$v(\vec{x}) = c_s A \sqrt{\left(\frac{\partial u}{\partial x}\right)^2 + \frac{1}{2}\left(\frac{\partial u}{\partial y} + \frac{\partial v}{\partial x}\right)^2 + \left(\frac{\partial v}{\partial y}\right)^2}, \tag{2}$$

where $A$ is the horizontal surface area of the grid cell where the particle is located, $u = u(\vec{x})$ and $v = v(\vec{x})$ are respectively the (depth dependent) zonal and meridional velocity components. As such, the magnitude of the stochastic noise depends on the local velocity field, and its variance increases linearly over the time that a particle is advected. The Smagorinsky viscosity depends strongly on the flow shear compared to other parameterisations [47].

The strength of the noise can be determined with the parameter $c_s \geq 0$. Multiple methods exist in LES to determine the value of $c_s$ in each application [48]. The velocity gradients $\left(\frac{v}{c_s A}\right)$ in the non-eddying version of POP typically range from $10^{-9} \mathrm{s}^{-1}$ to $10^{-7} \mathrm{s}^{-1}$ and $A \approx 10^4 \ \mathrm{km}^2$, so the estimated standard deviation of the zonal and meridional stochastic noise $(\hat{\sigma}_x(t), \ \hat{\sigma}_y(t))$ after 20 days ($\Delta t \approx 1.6 \cdot 10^6 \mathrm{s}$) range from $6\sqrt{c_s}$ km to $60\sqrt{c_s}$ km. These scales are similar to the mesoscale: 10-30 days and 10-100km for mesoscale eddies [49].

Altogether, we apply the particle tracking analysis in four different model configurations (see Table 1), and compare the distributions of particles at the ocean surface after the backtracking from a single release location; 130 particles are used at every release location to determine the particle distributions. These configurations represent the differences between state-of-the-art, global OGCM resolutions of the past (1° horizontally and monthly model output) and the present-day (0.1° horizontally and model output on a daily scale). The single effect of model output with lower temporal resolution compared to the state-of-the-art present-day OGCMs is investigated in a separate configuration $R_{0.1m}$.

We use three measures to compare the particle distributions between the configurations (see Fig 1b–1d): (i) the average lateral distance (km) travelled from the release location (along the red lines in Fig 1b), (ii) the surface area spanned by the particles approximated by the summed surface area of the 1° × 1° grid boxes (blue boxes in Fig 1c), and (iii) the Wasserstein distance $W_d$ as a measure of difference between two distributions resulting from two simulations. The Wasserstein distance is the minimum distance that one has to displace the particles resulting from one simulation (along the dashed lines in Fig 1d) to transform it into another particle distribution (and is calculated with [50]).

## Results

We first analyse the overall differences between the configurations $R_{0.1m}$, $R_{1m}$, $R_{1md}$ and the reference configuration $R_{0.1}$ in terms of the three measures described above (see Fig 1). Thereafter, we show specific release locations to explain why the configurations with lower spatial resolution do or do not provide similar solutions to the reference configuration $R_{0.1}$.

**Table 1. The configurations with simulations in varying OGCM resolutions.**

| Configuration | resolution | output | diffusion | remark |
|---|---|---|---|---|
| $R_{0.1}$ | 0.1° | once every 5 days | $c_s = 0$ | Reference case, $\vec{v}_b = 0$, eddying |
| $R_{0.1m}$ | 0.1° | monthly | $c_s = 0$ | $\vec{v}_b = \vec{0}$, eddying |
| $R_{1m}$ | 1° | monthly | $c_s = 0$ | non-eddying |
| $R_{1md}$ | 1° | monthly | $c_s \in [0.25, 0.5, 1.0, 1.5, 2.0, 5.0]$ | non-eddying |

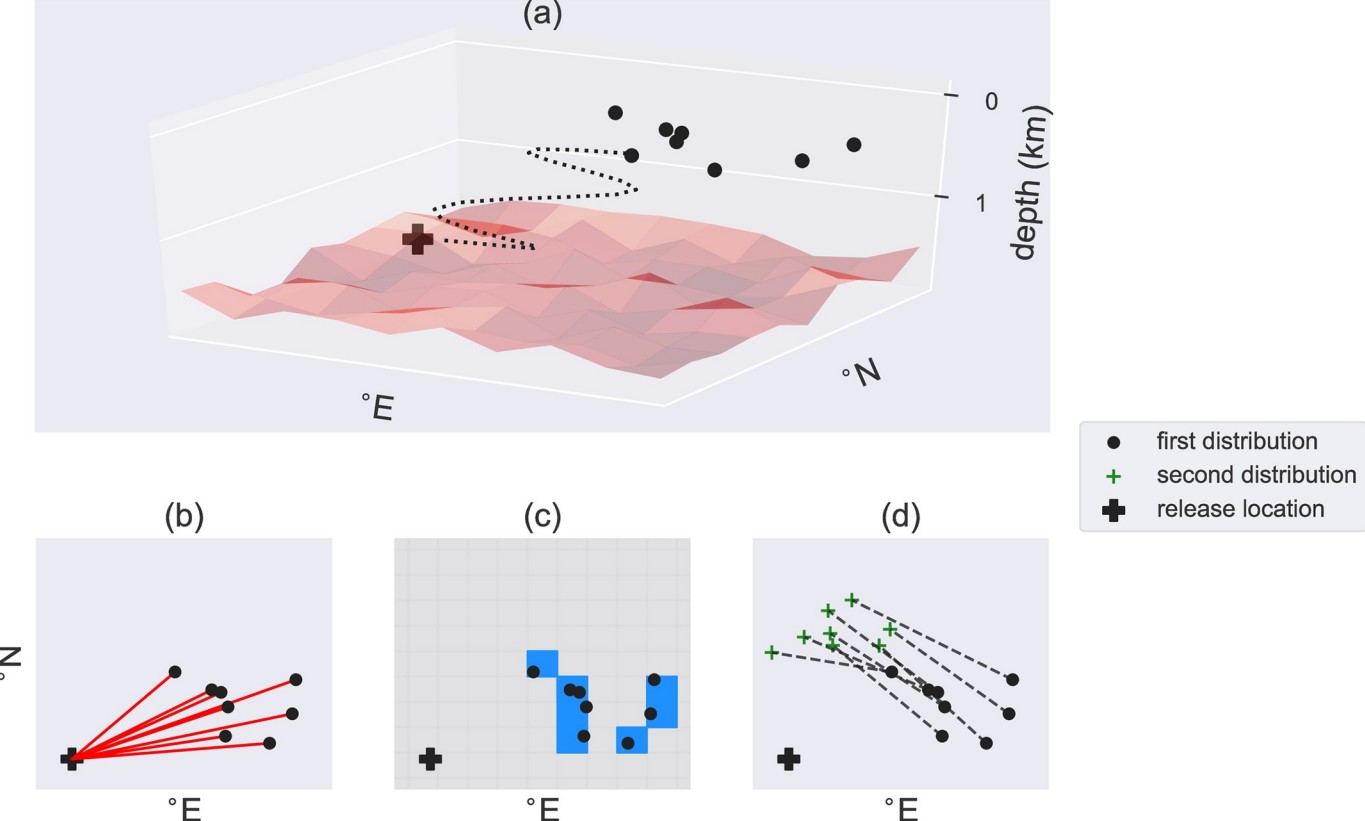

**Fig 1. Schematic illustration of (a) the back-tracking analysis and (b)-(d) the three measures which are used to compare the particle distributions at the ocean surface.** (a) Three-dimensional illustration: Particles are released at the bottom every three days for a period of around six years, back-tracked until they get close to the surface (10m depth), which results in a particle distribution at the surface. A map of (b) the average lateral distance (km) traveled from the release location (along the red lines), (c) the surface area (blue; km$^2$) spanned by the particle distribution (approximated by the summed surface area of the 1˚ × 1˚ blue boxes), (d) the Wasserstein distance ($W_d$; km), which is the minimum distance that one has to displace the particles (along the dashed lines) to transform one distribution into another distribution.

## Global analysis

The average lateral travel distances of the particles are globally different between the four configurations (Fig 2). In the configuration with lower spatial resolution $R_{1m}$, the average lateral displacement is more extreme compared to the reference configuration (i.e. it is larger in regions with relatively large displacement and lower in regions with low displacement; Fig 2c). The average travel distance in $R_{0.1m}$ is similar to the reference case $R_{0.1}$ (see Fig 2d for the global averages).

The average lateral displacement becomes globally less 'extreme' (especially the Southern Ocean peaks are lower) if the Smagorinsky diffusion is added to the flow dynamics in $R_{1md}$ compared to $R_{1m}$ (Fig 2c). The less extreme pattern of the travel distances explains why the globally averaged lateral travel distance is minimal at $c_s = 0.25$ (for $w_f = 6$ m day$^{-1}$ in Fig 2d), and not at $c_s = 0$. The coefficient $c_s$ influences the lateral displacement in two ways. First, more displacement is added per time step if the noise is stronger (for larger $c_s$), and the lateral displacement will on average be larger for larger $c_s$. Second, the noise will be larger in areas with strong flow ($u$ and $v$ in Eqs 1 and 2). Hence, for small $c_s$ the noise is large enough for the particles to travel outside of the areas with a relatively strong flow and large displacement (such as

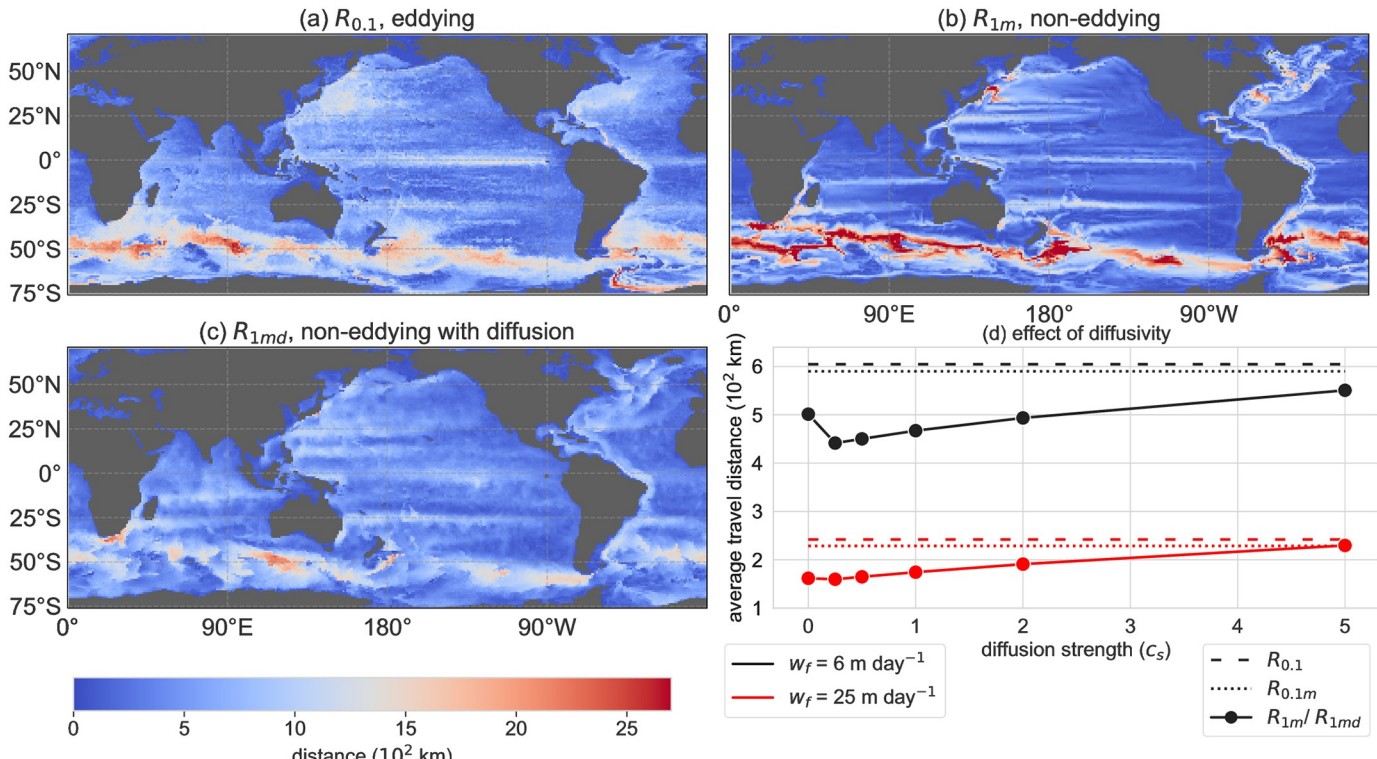

**Fig 2.** (a), (b), (c) The average horizontal distance between the release location and the final back-tracked location at the ocean surface with $w_f$ = 6 m day$^{-1}$ respectively in configuration $R_{0.1}$, $R_1$, $R_{1md}$ with diffusion strength $c_s$ = 2.0 (see Fig 4 for $R_{0.1m}$). (d) Global averaged lateral travel distance in all configurations (for several values of $c_s$ in $R_{1md}$). $w_f$ = 6 m day$^{-1}$ in black and $w_f$ = 25 m day$^{-1}$ in red.

in the Southern Ocean), such that the globally averaged lateral displacement is lower than for $c_s$ = 0.

The surface area spanned by the particle distributions (Fig 1b) is often smaller in $R_{0.1m}$ compared to $R_{0.1}$, as can be seen from the global average of this measure (Fig 3d). The lower surface area could be explained by the tendency of nearby particles to follow more similar pathways in $R_{0.1m}$ than in $R_{0.1}$ (see animation S1 Video) [25]. As a result, the particles will end up in clusters closer to each other at the surface. Hence, the surface area of the particle distributions is on average smaller in $R_{0.1m}$ compared to $R_{0.1}$.

Mesoscale eddies are abundant in the reference configuration $R_{0.1}$, while they are absent in the low spatial resolution configuration $R_{1m}$. Therefore, tracked particles tend to end up in a much more confined area at the surface in the lower resolution configuration $R_{1m}$ than in the reference configuration (Fig 3d). The stochastic noise in $R_{1md}$ induces fluctuations in the particle trajectories, leading to a larger surface area of the particle distributions. In $R_{1md}$, the global average surface area of the particle distributions increases monotonically with increasing magnitude of the noise ($c_s$).

Interestingly, the value of $c_s$ that approximates configuration $R_{0.1}$ ($c_s \approx 3.5$) and $R_{0.1m}$ ($c_s \approx$ 2) best, is the same for both sinking velocities 6, 25 m day$^{-1}$. These values of $c_s$ must result in a similar scale of the flow fluctuations ($\sigma_x$, $\sigma_y$) in configurations $R_{0.1}$ and $R_{0.1m}$. It also indicates that, given $c_s$, the subgrid-scale parameterisation performance is similar for both sinking velocities. Nevertheless, the particle distributions match better with the reference configuration if the sinking velocity is higher (according to the $W_d$ in Fig 2), because a lower particle travel time leads to less spread of the particle trajectories and a lower lateral displacement.

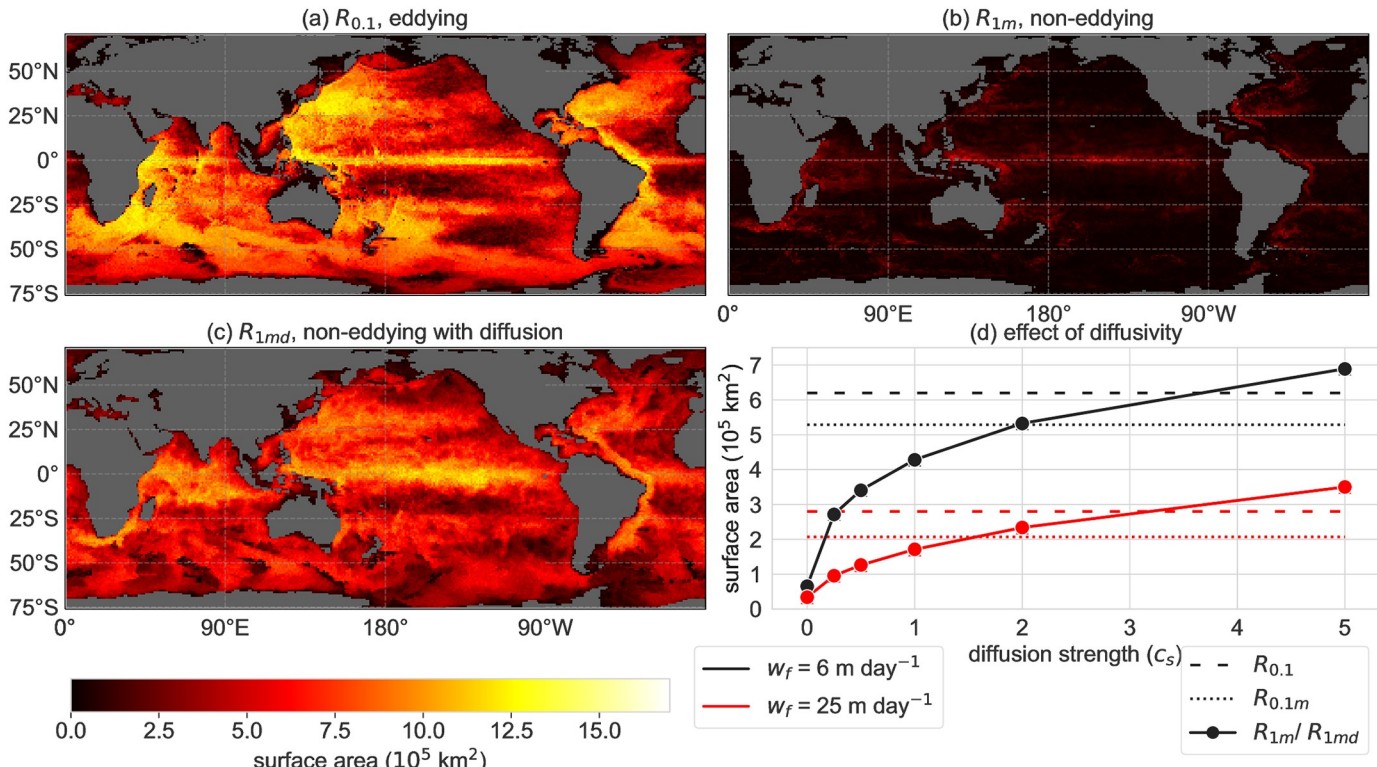

**Fig 3.** (a), (b), (c) The surface area of the back-tracked particle distributions with $w_f = 6$ m day$^{-1}$ respectively in configuration $R_{0.1}$, $R_1$, $R_{1md}$ with diffusion strength $c_s = 2.0$ (see Fig 5 for $R_{0.1m}$). (d) Globally averaged surface area of the particle distributions in all configurations (for several values of $c_s$ in $R_{1md}$). $w_f = 6$ m day$^{-1}$ in black and $w_f = 25$ m day$^{-1}$ in red.

Locally, the surface area of the particle distributions shows a different pattern in $R_{1md}$ compared to the reference $R_{0.1}$ (Fig 3a vs Fig 3c). In contrast to the magnitude of the noise, the direction of the noise vector does not depend on the flow field (it is horizontally isotropic). Therefore, the surface area of the particle distributions in configuration $R_{1md}$ is overestimated in the tropics compared to the reference configuration $R_{0.1}$, where the flow is mostly zonal. Interestingly, this measure remains low in areas with sinking waters for both configurations $R_{0.1}$ and $R_{0.1md}$, such as the Ross sea and the Weddell sea (see Fig 2 in [51]).

The loss of information in $R_{0.1m}$ due to the monthly averaging of the flow fields in $R_{0.1}$ is clearer in the difference plots of the surface area and travel distance of the particle distributions (Fig 4). The surface area of the particle distributions is mostly lower in $R_{0.1m}$ compared to $R_{0.1}$ (Fig 4a). The particles tend to be advected by a similar flow field in $R_{0.1m}$ if they are located close to each other. Hence, groups of particles are trapped in the same eddies, and travel from origin locations at the ocean surface which are closer to each other. This could result in notably different back-tracked particle distributions, especially if the shear of the flow field is high (see for instance the location 45.5˚S, 39.5˚E on planktondrift.science.uu.nl or S1 Fig for two similar locations with opposite behaviour).

In general, we find that a reduction of the temporal resolution ($R_{0.1m}$ vs. the reference case $R_{0.1}$) does not have a major effect on the Wasserstein distance $W_d$ (Fig 5). The travel time of the particles is perhaps too short (at most a few years) for the errors in $R_{0.1m}$ to grow substantially, and remains smaller compared to $R_{1m}$. The global average $W_d$ between $R_{0.1m}$ and the reference case ($W_d(R_{0.1}, R_{0.1m})$) is slightly larger compared to the check of $R_{0.1}$ with itself (the

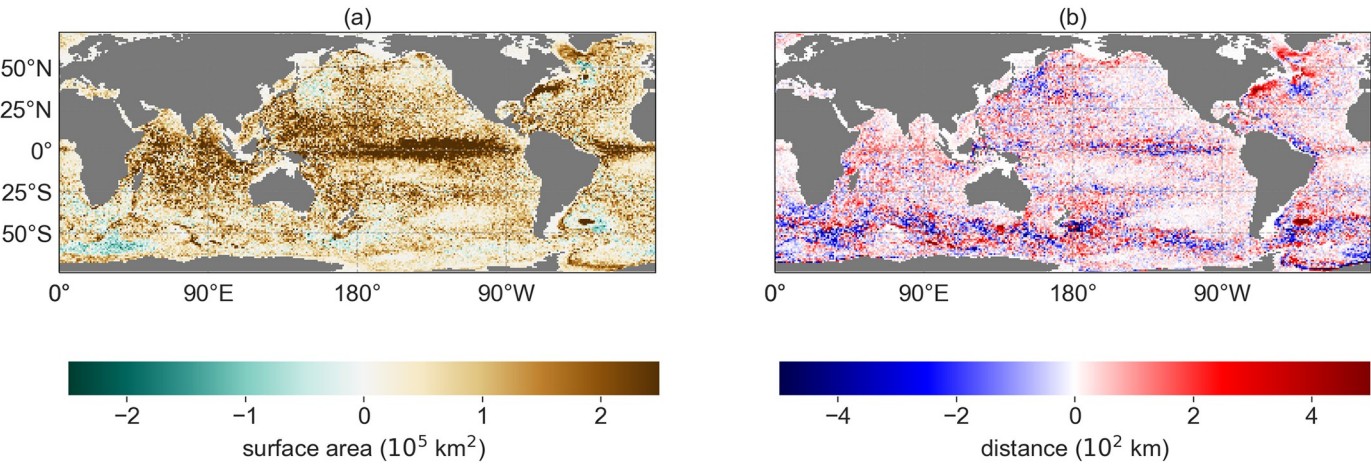

**Fig 4. The differences between $R_{0.1m}$ and $R_{0.1}$ ($R_{0.1m}$ subtracted from $R_{0.1}$) in terms of (a) the surface area of the particle distributions (Fig 1c) and (b) the average travel distances of the particle distributions (Fig 1b).**

global average $W_d(R_{0.1}, R_{0.1})$; dashed versus dotted in Fig 5d). In this 'check', we did the same analysis as in $R_{0.1}$, but with a 1.5 day shift of the particle release times. As a result, the particle distributions will be different in the check, but as similar as one could get to the particle distributions of $R_{0.1}$ in the other configurations.

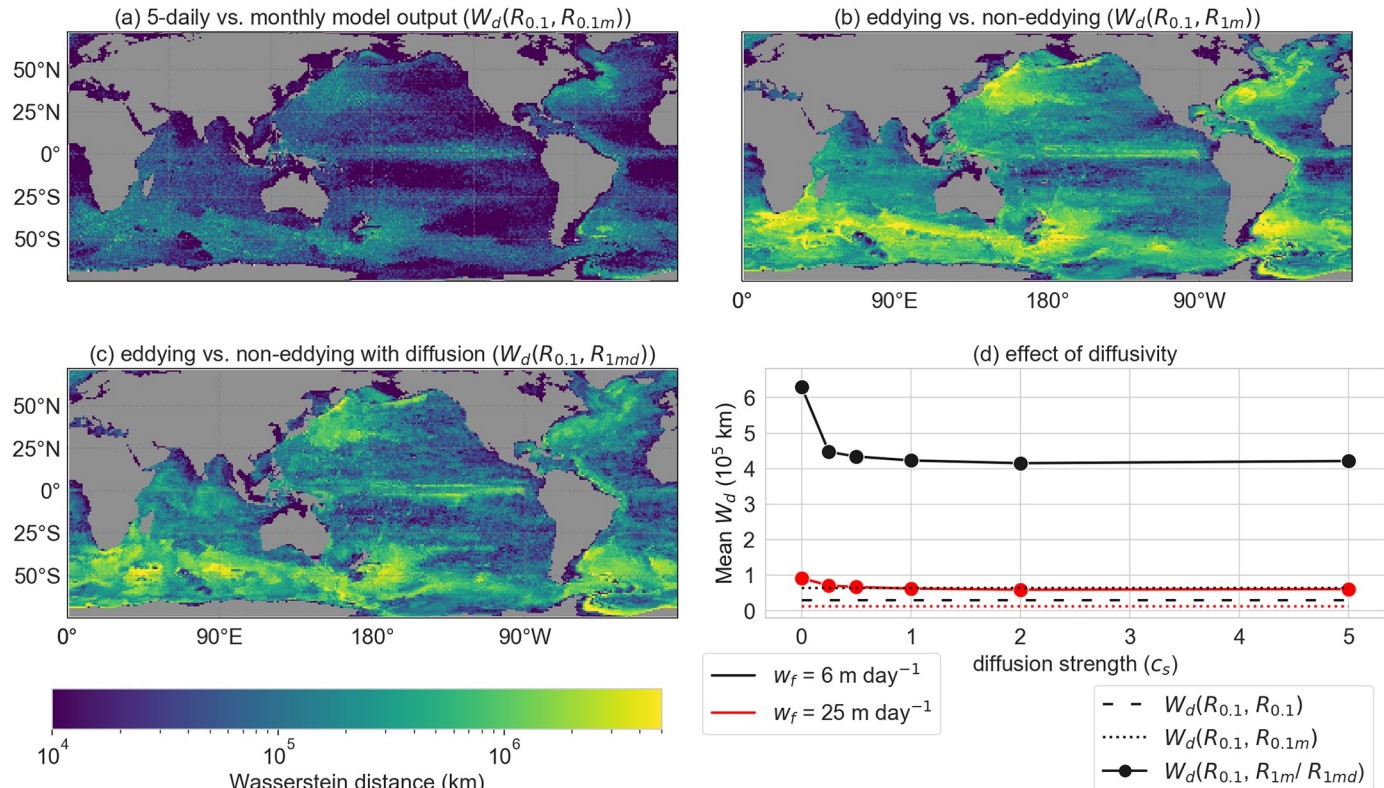

**Fig 5. The global Wasserstein distance ($W_d$) as a distance measure between the back-tracked particle distributions of the configurations from Table 1.** $W_d$ for sinking speed $w_f = 6$ m day$^{-1}$ between the eddying reference configuration $R_{0.1}$ and (a) the eddying $R_{0.1m}$ with monthly model output, (b) the non-eddying $R_{1m}$, (c) the non-eddying $R_{1md}$ with diffusion strength $c_s = 2$. (d) Global average $W_d$ in all configurations (for several diffusion strengths $c_s$ in $R_{1md}$). $w_f = 6$ m day$^{-1}$ in black and $w_f = 25$ m day$^{-1}$ in red. The globally averaged $W_d$ of configuration $R_{0.1}$ with itself is a check ($W_d(R_{0.1}, R_{0.1})$; only for $w_f = 6$ m day$^{-1}$), as the globally averaged $W_d$ is shown between the particle distribution of the same configuration, but with a 1.5-day shift of the particle release times.

On the other hand, altering the spatial resolution in $R_{1m}$ and $R_{1md}$ does lead to different values of the $W_d$. We find that any value of $c_s > 0$ reduces the $W_d$ by a similar amount, but the $W_d$ is the smallest for $c_s = 2$ with both sinking speeds $w_f = 6, 25$ m day$^{-1}$ (Fig 5d). For $c_s =$ 2, the approximated zonal and meridional standard deviation of the diffusion ($\hat{\sigma}_x(t), \hat{\sigma}_y(t)$) ranges between 8km and 80km in 20 days (depending on the strength of the local velocity gradients in the model). At this value of $c_s$, the magnitude of the fluctuations from the eddies lead to the optimal parameterisation, such that the particle distributions spread enough to better match with the particle distributions in the reference case. However, we find that the global averaged $W_d$ for $c_s = 2$ is approximately eight times larger compared to the check ($W_d(R_{0.1}, R_{0.1})$) for $w_f = 6$ m day$^{-1}$, which implies that the particle distributions differ substantially from the reference case. In all configurations $R$, $W_d(R_{0.1}, R)$ is lower in areas where the divergence of particle trajectories is relatively small, such as areas of relatively low eddy kinetic energy (e.g. in the gyres; see supporting information), and in areas where the travel time of the particles is relatively short because of the shallow bathymetry (or the particles sink faster).

## Regional analysis

In general, the particle trajectories in the lower spatial resolution configuration $R_{1m}$ without diffusion are different compared to the trajectories in the reference configuration $R_{0.1}$, because these trajectories lack the fluctuations provided by eddies and hence they spread less. The only trajectory spread in the non-eddying $R_{1m}$ is caused by flow variability on a larger timescale, such as seasonality. We focus here on some specific locations to see how this can lead to different particle distributions.

If Smagorinsky diffusion is added to the dynamics of the flow ($R_{1md}$), the fluctuations from the eddies are parameterised and the trajectories spread more. The North Pacific gyre is a location where this parameterisation works well (Fig 6a). Within the gyre, the diffusion is relatively low in the reference configuration and the eddies spread the particle trajectories uniformly in all directions. Adding fluctuations to the flow field in $R_{1md}$ using stochastic noise captures the spread of these eddies in $R_{0.1}$ well. Occasionally the parameterisation also works well in locations with larger shear and eddy activity compared to the North Pacific gyre. For example, for a location in the Antarctic Circumpolar Current (ACC, Fig 6b), the mean flow field (averaged over 6 years) in $R_{1m}$ is similar to the mean flow field in $R_{0.1}$. The stochastic noise can again adequately capture the effect of fluctuations provided by the eddies on the particle distributions.

However, it is well known that non-eddying OGCMs do not get the mean flow field right in all of the locations, because the eddies influence the mean flow field through rectification [52]. The Agulhas region is such an example where the mean flow field is different in $R_{1m}$ compared to the reference case $R_{0.1}$ (Fig 6c). The analysis in $R_{1md}$ provides a particle distribution which only comprises a subset of the particle distribution from the analysis in $R_{0.1}$. If the strength of the noise ($c_s$) is increased here, at most the spread of the particle distribution increases, but one will not find that any particle originates from the area around Madagascar.

Finally, the addition of spatially dependent noise has one more unrealistic property: The particles tend to artificially accumulate in areas with relatively low horizontal gradients, and hence weak stochastic noise [53–55]. A result of this effect can be found in another location near the ACC, South of Australia (Fig 6d). At this location, configuration $R_{1md}$ results in two clusters of particles, which are separated by an area with high shear, and where the noise is large, while the particles in the reference configuration $R_{0.1}$ clearly form one (more connected) distribution.

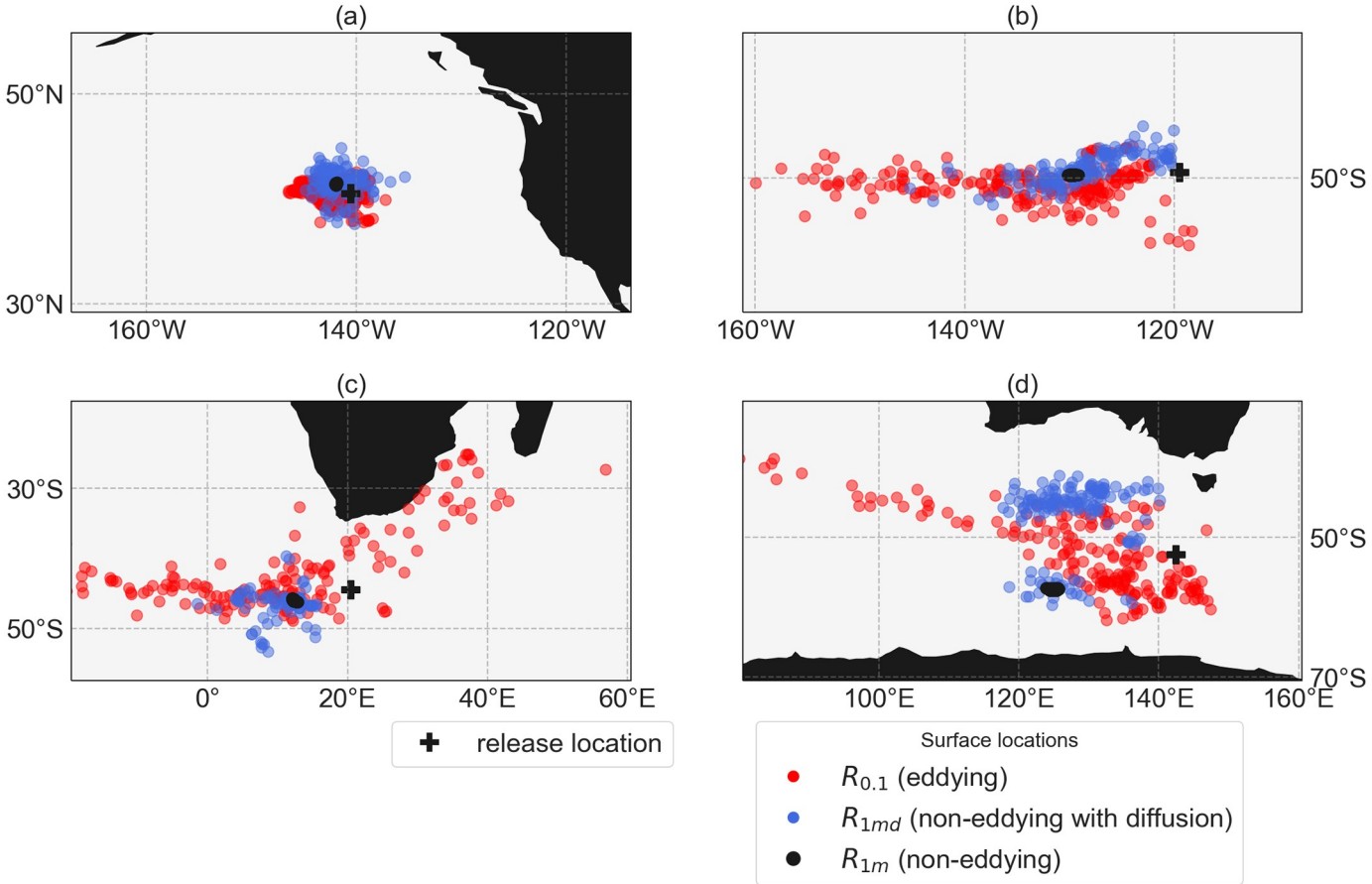

**Fig 6. Comparison between reference configuration $R_{0.1}$ (red), configurations $R_{1m}$ (yellow) and $R_{1md}$ with $c_s$ = 2 (blue) at four different locations (yellow on top of blue, blue on top of red; $w_f$ = 6 m day$^{-1}$). Each distribution consists of $\sim$ 160 particles.** The release locations at (a) 40.5˚N, 140.5˚W and respectively 4601 and 4542 m depth in $R_{0.1}$ and $R_{1m}$, (b) 49.5˚S, 119.5˚W and respectively 3122 and 3249 m depth in $R_{0.1}$ and $R_{1m}$, (c) 44.5˚S, 20.5˚E and respectively 4249 and 4249 m depth in $R_{0.1}$ and $R_{1m}$, (d) 52.5˚S, 142.5˚E and respectively 3070 and 2916 m depth in $R_{0.1}$ and $R_{1m}$.

## Discussion

We assessed the variations of Lagrangian trajectories of sinking particles in flow fields which were generated by OGCMs of different resolutions. We released sinking particles at the bottom of the ocean, tracked them backwards in time until they reached the surface, and investigated how the particle distributions at the ocean surface depend on the OGCM resolution.

If the model output of the high-resolution OGCM is averaged from 5-daily to monthly data, the particle tracking analysis provides similar results in most cases. However, in some specific regions with large shear, we find notable differences of the back-tracked particle distributions at the ocean surface.

Overall, the sinking Lagrangian particles give unrealistic results in the non-eddying models, because (1) the back-tracked distributions show too little spread due to the absence of ocean eddies and (2) these models often do not capture the mean flow fields correctly (as shown in e.g. [29, 30, 33]). Lateral stochastic diffusion in the low-resolution configuration (re-)introduced part of the eddy fluctuations and hence increased the relative dispersion of particle trajectories, and increased the lateral travel distance (Fig 2) and the spread (Fig 3) of the back-tracked particle distributions. Hence this method is promising for locations where the low

resolution OGCMs capture the mean flow field well. However, the particle distributions are often distant from the distributions of the reference configuration, as is shown from the Wasserstein distance in Fig 5, which implies that the surface origin location is different between the coarse and high resolution models. Therefore, overall the Smagorinsky diffusion is insufficient to parameterise the eddies in most areas.

Altogether, we recommend to compute the sinking Lagrangian particle trajectories only in eddying OGCMs. We used the Smagorinsky parameterisation in this paper as a first attempt to represent the subgrid-scale processes if the eddies are absent in the flow. Other types of parameterisations could be applied. Several other parameterisations for eddy-induced mixing of tracers are available in POP [56]. However, the improvement of either the Eulerian or Lagrangian parameterisation of the subgrid scale variability in the flow remains a challenge in ocean modelling [57].

These conclusions have implications for Lagrangian particles in paleoceanographic models. OGCMs used in most paleo studies lack the eddying flow characteristics and do not generate a locally representative time mean flow for the time period of interest. Since Lagrangian particles use the local flow field, they require eddying paleoceanographic models that better represent the time mean flow for the considered time period. For the application of Lagrangian particle tracking techniques in paleoceanographic models, which are usually not eddying, we recommend to test model results first against independent information of ocean flow, such as biogeographic patterns of microplankton [58, 59]. In turn, it should be appreciated that a regional paleoceanographic signal could be influenced by flow characteristics which are not represented by the non-eddying models. This represents a cautionary tale in putting too much confidence in flow fields from low-resolution fully coupled GCM simulations.

Future work could investigate the sinking Lagrangian particles in other configurations with different OGCM resolutions. The model output of configuration $R_{0.1}$ could be coarsened to a 1˚ grid before the back-tracking analysis, to separate out the effects on the Lagrangian analysis of (a) a coarsened grid (see also [60, 61]) and (b) a lower resolution of the underlying Eulerian model. Moreover, particle trajectories could be sensitive to the vertical resolution of the OGCM (e.g. [62, 63]).

When models do not resolve the so-called internal Rossby deformation radius (about 50 km at midlatitudes), no eddies can be represented. When the model grid scale is only slightly smaller than the deformation radius, say 25 km at midlatitudes, eddies form but their interaction is not fully captured; such a model is called 'eddy permitting.' Only for models at about 1 km horizontal midlatitude resolution (so-called eddy-resolving models), eddy interactions are fully resolved. The 10 km resolution POP model, as is used here ($R_{0.1}$), is therefore often called 'strongly eddying.' An OGCM of ∼1km resolution also has a better representation of the spatial/temporal submesoscale that can be important for sinking Lagrangian particles which represent the carbon flux to the ocean bottom [64]. Although the mesoscale flow contains most of the energy that is responsible for the tracer dispersion [65], submesoscale (1-20km) dynamics have proven to be of importance for the vertical advection of iron in specific regions with strong flow-bathymetric interactions [66]. Future work could analyse the transport of sinking particles in models with higher resolutions, and with models which better represent internal tides [67] or improved interaction of the bottom-flow with topography [68].

The effect of eddies on the flow should be appreciated outside the physical oceanography community. In order to facilitate increased understanding on this matter, interactively disseminate our results, and allow users to self-explore and verify the surface-ocean location of origin for sedimentary particles, we developed the website planktondrift.science.uu.nl containing our results.

We also tested an additional configuration without the bolus velocity in the non-eddying POP model (i.e. the same as $R_{1md}$ and $R_{1m}$, but where $v_b = \vec{0}$). The results for this configuration are very similar to the results that are obtained in configuration $R_{1md}$ and $R_{1m}$ in this paper. The bolus velocity is weaker compared to the Eulerian flow velocity (typically $v_b$ is approximately 5% of $v_a$ at the surface layer). Parameterisations like GM improve the temperature and salinity distribution in Eulerian models. GM is a type of 'extra advection' which assumes that dynamic tracers such as temperature and salinity mix along surfaces of contstant potential density [35]. However, GM does not make a relevant difference if Lagrangian particles are applied offline to represent other tracers. The results for this additional configuration can be found and downloaded from the planktondrift.science.uu.nl website.

The website contains the results which are presented in this paper, for every release location in every configuration. Users of the tool can choose a location at the bottom of the ocean, see where the sinking particles originated from for different parameters (e.g. the sinking speed $w_f$, or the magnitude of the noise $c_s$), and download these origin locations. The website allows anyone who works with e.g. sedimentary microplankton assemblages or plastic to see how the sinking particles could be displaced laterally, and what the environment (e.g. sea surface temperature and salinity) is at the displaced location using POP or other OGCMs. Hence, the advection bias [17] of the sedimentary assemblages can be determined in the present-day ocean.

## Supporting information

**S1 Fig. Comparison between the reference configuration $R_{0.1}$ (red) and the temporally averaged configuration $R_{0.1m}$ (blue) at two release locations ($w_f$ = 6 m day$^{-1}$).** (a) 45.5˚S, 39.5˚E at 2068m depth (red on top of blue) (b) 46.5˚S, 42.5˚E at 2238m depth (blue on top of red).
(TIF)

**S2 Fig. Geographic plot of the time mean eddy kinetic energy at the surface.** The eddy kinetic energy is defined as $\frac{1}{2} \overline{u' \cdot u'}$, where the bar denotes the time mean and $u'$ the deviation from the time mean velocity vector $u$ (so $u(\vec{x}, t) = \bar{u}(\vec{x}) + u'(\vec{x}, t)$).
(TIF)

**S1 Video. Animation (back in time) of particle back-tracking analysis ($w_f$ = 6 m day$^{-1}$) with particle release at the Uruguayan margin (47.9˚E and 37.15˚S, $\sim$4800m depth).** (a) the configuration $R_{0.1}$ with 5-daily model ouput and (b) the configuration $R_{0.1m}$ with monthly model output.
(MOV)

## Acknowledgments

The code used for this work and the results are distributed under the MIT license and can be found at the website https://github.com/pdnooteboom/PO_res_error. PN thanks Jasper de Jong and Daan Reijnders for their help with the implementation of the Smagorinsky parameterisation in Parcels.

## Author Contributions

**Conceptualization:** Peter D. Nooteboom, Philippe Delandmeter, Erik van Sebille, Peter K. Bijl, Henk A. Dijkstra, Anna S. von der Heydt.

**Data curation:** Peter D. Nooteboom.

**Formal analysis:** Peter D. Nooteboom.

**Funding acquisition:** Erik van Sebille, Peter K. Bijl, Henk A. Dijkstra, Anna S. von der Heydt.

**Investigation:** Peter D. Nooteboom.

**Methodology:** Peter D. Nooteboom, Philippe Delandmeter.

**Project administration:** Peter D. Nooteboom.

**Resources:** Henk A. Dijkstra, Anna S. von der Heydt.

**Software:** Peter D. Nooteboom, Philippe Delandmeter, Erik van Sebille.

**Supervision:** Erik van Sebille, Peter K. Bijl, Henk A. Dijkstra, Anna S. von der Heydt.

**Validation:** Peter D. Nooteboom.

**Visualization:** Peter D. Nooteboom.

**Writing – original draft:** Peter D. Nooteboom.

**Writing – review & editing:** Peter D. Nooteboom, Philippe Delandmeter, Erik van Sebille, Peter K. Bijl, Henk A. Dijkstra, Anna S. von der Heydt.

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
