## [Decision Letter · Decision Letter 0]

27 Apr 2020

PONE-D-20-08829

Resolution-dependent variations of sinking Lagrangian particles in general circulation models

PLOS ONE

Dear Mr Nooteboom,

Thank you for submitting your manuscript to PLOS ONE. After careful consideration, we feel that it has merit but does not fully meet PLOS ONE’s publication criteria as it currently stands. Therefore, we invite you to submit a revised version of the manuscript that addresses the points raised during the review process. It is particularly important for the authors to identify more clearly the audience they target with this work, and that they carefully revise the manuscript accordingly. Please read carefully the suggestions made by the reviewers so that they are addressed in the revised version.

We would appreciate receiving your revised manuscript by Jun 11 2020 11:59PM. To enhance the reproducibility of your results, we recommend that if applicable you deposit your laboratory protocols in protocols.io, where a protocol can be assigned its own identifier (DOI) such that it can be cited independently in the future. For instructions see: http://journals.plos.org/plosone/s/submission-guidelines#loc-laboratory-protocols

We look forward to receiving your revised manuscript.

Kind regards,

Vanesa Magar, Ph.D.

Academic Editor

PLOS ONE

Journal Requirements:

2. We note that Figures 2, 3, 4, 5, 6 and S1  in your submission contain [map/satellite] images which may be copyrighted. All PLOS content is published under the Creative Commons Attribution License (CC BY 4.0), which means that the manuscript, images, and Supporting Information files will be freely available online, and any third party is permitted to access, download, copy, distribute, and use these materials in any way, even commercially, with proper attribution. For these reasons, we cannot publish previously copyrighted maps or satellite images created using proprietary data, such as Google software (Google Maps, Street View, and Earth). For more information, see our copyright guidelines: http://journals.plos.org/plosone/s/licenses-and-copyright.

a)    You may seek permission from the original copyright holder of Figures 2, 3, 4, 5, 6 and S1 to publish the content specifically under the CC BY 4.0 license.  

Reviewers' comments:

Reviewer's Responses to Questions

**Comments to the Author**

1. Is the manuscript technically sound, and do the data support the conclusions?

Reviewer #1: Yes

Reviewer #2: Partly

Reviewer #3: Partly

2. Has the statistical analysis been performed appropriately and rigorously? 

Reviewer #1: Yes

Reviewer #2: Yes

Reviewer #3: Yes

3. Have the authors made all data underlying the findings in their manuscript fully available?

Reviewer #1: Yes

Reviewer #2: Yes

Reviewer #3: Yes

4. Is the manuscript presented in an intelligible fashion and written in standard English?

Reviewer #1: Yes

Reviewer #2: Yes

Reviewer #3: Yes

5. Review Comments to the Author

Reviewer #1: This paper is a numerical discussion of particle motion in ocean general circulation models, principally as a function of spatial resolution. The conclusions are unsurprising: eddy resolution makes a big difference to long-term particle displacements. My inference is that the paper is directed *not* at numerical modelers, but at paleoclimate people who over-interpret results of crude circulation models. If this inference is correct, it should be made explicit in the abstract, introduction, and probably the title and raises the question of whether it belongs in a paleo-journal.

If the intended audience is not the technically adept numerical modelling community, it is essential to tell the reader that the paper represents errors in Lagrangian displacements in *Eulerian* models. Non-expert readers should be told that although not common, Lagrangian circulation models exist, as do whole books on the subject (e.g., Bennett's). An interesting, technical, question would be how well a Lagrangian GCM can calculate the Eulerian mean flows.)

Why is a backwards in time calculation done? Does some stability issue not arise?

Again, for non-specialist audiences, it would be helpful to point out that skill in calculating the distribution of scalar tracers, such as temperature might be much more accurate in the parameterized Eulerian models---given how the parameterizations are constructed.

Reviewer #2: The manuscript describes how statistics of Lagrangian sinking particles differ when advected off-line in an eddy permitting or coarse resolution ocean model. While realizing the need for the work presented for some communities (paleo?), from a physical oceanography point of view, the content is not novel and mostly qualitative. It is well known that ocean models that are not eddy resolving underestimate horizontal dispersion. The literature on the subject is very large and some references should indeed be added. I note also that the manuscript in its current form reads like a physical oceanography work, and the appeal to the broader and more biologically oriented community of PLOS One is unclear to me.

I do not have major comments on what is in the manuscript (there is nothing wrong per se), but I would have hoped for a more in-depth justification of the work and its limitations considering the lack of novelty, or a more in-depth investigation of the reliability of the high and low resolution results. The outcome of this work confirms deductions that have been made already in the absence of sinking, and many Lagrangian studies have been performed in this regard comparing runs at the exact resolutions used here (to mention few relevant to the content of this manuscript: Putman and He, 2013; Doos, Rupolo and Brodeau, 2011, and many intercomparison papers focused on the Eulerian fields).

In detail:

Firstly, the manuscript never mentions the resolution limitations of the simulation used. 10km is not enough to resolve eddies at high latitudes, therefore the run is eddy permitting. No mention is made of the scales that are not resolved (effectively 30 km and below) in the high resolution case. Nothing is said about the vertical resolution (work by Stewart et al, 2017 and Bracco et al., 2018, both in Ocean Modelling may be of interest). In other words, can we expect to see similar results if we had a 1km resolution ocean model or 100 vertical layers? Accounting for work by, for example, Patrice Klein and others, the likely answer is no. Global runs at 2km have been made and are potentially freely available for analysis (MIT model, OFES…).

Secondly, what would happen is a reanalysis data-set was used instead? The most recent ones are run at 25km->10km near the poles and fields are saved monthly or more frequently. Do the statistics in the 10km run compare well with those from a reanalysis product, where at least surface eddies are constrained by satellite observations and the water column dynamics assimilate the large ARGO dataset? The investigation of the time-dependence of the results (advecting particles using fields saved every month or every 5 days is also superficial and should be expanded).

Thirdly, I understand the argument for justifying the simulation done for small plastic particles, but as far as I know organic particles small enough to have a sinking velocity of 6-20 m/day near the surface do not make it to the ocean bottom as such (vice versa particles that may reach the bottom with these sinking velocities were likely much larger and heavier at formation). Aggregation, grazing and bacterial activities are dominant processes on those sizes. The review by Boyd et al. in Nature in 2019 has a nice image summarizing the fate of sinking particles as function of their size, and shows that sinking is not the dominant process determining the fate of particles as small as those used in the manuscript (they do not reach more than 400-600 m depth in particle form). It is unclear, therefore, what is the paleo relevance that justifies the need for coarse resolution ocean models.

Overall, in my opinion this is an exercise that deserves publication if better motivated, with a more in-depth discussion of implications and limitations, and the exploration of a greater range of sinking velocities, saving times, etc.

Reviewer #3: This manuscript considers the problem of identifying the source location of sinking particles found at the bottom of the ocean and examines the effect of coarse spatial (and to some extent temporal) resolution. The authors find that ocean general circulation models (OGCMs) at a resolution typical of state-of-the-art paleoclimate applications (1 degree) do not capture the distributions of potential source locations well. This can be remedied to some degree by using a Smagorinski parameterisation for the eddy effects that are not directly modeled.

I find the experiment reasonably designed and the manuscript overall well written, although the organization is a bit odd. Some of the discussion could also benefit from a broader perspective, considering alternative explanations, and some of the conclusions are stated too strongly. Therefore, I recommend this submission for publication subject to minor revisions, which should address the detailed comments provided in the attachment.

6. PLOS authors have the option to publish the peer review history of their article (what does this mean?). If published, this will include your full peer review and any attached files.

Reviewer #1: No

Reviewer #2: No

Reviewer #3: No

---

## [Author Response · Author response to Decision Letter 0]

11 Jun 2020

Please see the submitted file 'Response_to_reviewers.pdf' to find our response to the editor and the reviewers, and the changes in the manuscript.

---

## [Decision Letter · Decision Letter 1]

4 Jul 2020

PONE-D-20-08829R1

Resolution dependency of sinking Lagrangian particles in ocean general circulation models

PLOS ONE

Dear Dr. Nooteboom,

Thank you for submitting your manuscript to PLOS ONE. After careful consideration, and based on the comments by the two reviewers, we feel that it has merit but does not fully meet PLOS ONE’s publication criteria as it currently stands. Therefore, we invite you to submit a revised version of the manuscript that addresses the points raised by the reviewers during the review process.

We look forward to receiving your revised manuscript.

Kind regards,

Vanesa Magar, Ph.D.

Academic Editor

PLOS ONE

Reviewers' comments:

Reviewer's Responses to Questions

**Comments to the Author**

1. If the authors have adequately addressed your comments raised in a previous round of review and you feel that this manuscript is now acceptable for publication, you may indicate that here to bypass the “Comments to the Author” section, enter your conflict of interest statement in the “Confidential to Editor” section, and submit your "Accept" recommendation.

Reviewer #1: (No Response)

Reviewer #3: All comments have been addressed

2. Is the manuscript technically sound, and do the data support the conclusions?

Reviewer #1: Yes

Reviewer #3: Yes

3. Has the statistical analysis been performed appropriately and rigorously? 

Reviewer #1: Yes

Reviewer #3: Yes

4. Have the authors made all data underlying the findings in their manuscript fully available?

Reviewer #1: Yes

Reviewer #3: Yes

5. Is the manuscript presented in an intelligible fashion and written in standard English?

Reviewer #1: Yes

Reviewer #3: Yes

6. Review Comments to the Author

Reviewer #1: This revised ms. is much improved in making clear what is the goal, and how it has been addressed. The paper is now clearly addressed to what are expected to be naive paleoceanographers who take model results much too seriously. As such, I would expect the authors to have sent this paper to some place like Quaternary Science Reviews, but that is their business (and that of the Editor of PLOS ONE).

Assuming everyone agrees that the journal is appropriate, the paper can be accepted in principle, but there exist a number of amplifications that would be very useful if the readership really is thought to be uninformed about numerical ocean models.

Readers need to be told that physical oceanographers distinguish between "eddy-permitting" models and "eddy-resolving" ones. This distinction would be crucial to anyone taking the paper seriously,  but it is not even mentioned. Are internal tides, internal waves, sub-mesoscales, of no concern?

The Wasserstein distance is fundamental to their results, but the paper has no explanation except for a reference number. Explain. 

Even in simple laminar flows, particle trajectories are known to be chaotic (e.g., Regier and Stommel, 1979 PNAS). Why is that not an issue here? 

It is likely true that the vertical velocity in the ambient fluid is negligible compared to the sinking velocities.  But in upwelling zones? (See e.g., Liang JGR 2018) Again, useful information for non-modelers. 

The complexity now known about near-bottom flows in the presence of topography needs to be at least mentioned. A large recent literature on this subject and probably very important in the particles landing in sedimentary regions.

The authors keep referring to the "mean" flow. Is there a definable mean flow? Over what time-scale? Same everywhere?

The bolus velocity is mentioned on P. 14. If it was ever defined, I missed it.

Can something be said about the skill of the reference model in reproducing the modern circulation elements, such as the eddy intensities, spatial scales, etc.?  

Consider comparing Fig. 3 to the "ventilation" Fig. 2 of Gebbie and Huybers (GRL 2011). Just needs one sentence.

Is "rectification" (P. 7) the right word? Comes from radio engineering and implies self-interaction producing a mean or low frequency.

Put  horizontal dimensions scales on Fig. 1.

Reviewer #3: The authors have done a good job addressing my previous concerns. I now recommend the manuscript for publication.

7. PLOS authors have the option to publish the peer review history of their article (what does this mean?). If published, this will include your full peer review and any attached files.

Reviewer #1: No

Reviewer #3: No

---

## [Author Response · Author response to Decision Letter 1]

6 Aug 2020

Please see the 'reviewer_response.pdf' file for the response to the reviewer comments.

---

## [Decision Letter · Decision Letter 2]

21 Aug 2020

Resolution dependency of sinking Lagrangian particles in ocean general circulation models

PONE-D-20-08829R2

Dear Dr. Nooteboom,

We’re pleased to inform you that your manuscript has been judged scientifically suitable for publication and will be formally accepted for publication once it meets all outstanding technical requirements.

Kind regards,

Vanesa Magar, Ph.D.

Academic Editor

PLOS ONE

Additional Editor Comments (optional):

Reviewers' comments:

Reviewer's Responses to Questions

**Comments to the Author**

1. If the authors have adequately addressed your comments raised in a previous round of review and you feel that this manuscript is now acceptable for publication, you may indicate that here to bypass the “Comments to the Author” section, enter your conflict of interest statement in the “Confidential to Editor” section, and submit your "Accept" recommendation.

Reviewer #1: All comments have been addressed

2. Is the manuscript technically sound, and do the data support the conclusions?

Reviewer #1: Yes

3. Has the statistical analysis been performed appropriately and rigorously? 

Reviewer #1: Yes

4. Have the authors made all data underlying the findings in their manuscript fully available?

Reviewer #1: Yes

5. Is the manuscript presented in an intelligible fashion and written in standard English?

Reviewer #1: Yes

6. Review Comments to the Author

Reviewer #1: I thought the previous comments were sufficiently minor that I didn't have to see this 3rd version Nonetheless,I have read it again. The authors do seem to have met the most important criticisms and the paper should beaccepted.

7. PLOS authors have the option to publish the peer review history of their article (what does this mean?). If published, this will include your full peer review and any attached files.

Reviewer #1: No

---

## [Editor Report · Acceptance letter]

25 Aug 2020

PONE-D-20-08829R2 

Resolution dependency of sinking Lagrangian particles in ocean general circulation models 

Dear Dr. Nooteboom:

I'm pleased to inform you that your manuscript has been deemed suitable for publication in PLOS ONE. Congratulations! Your manuscript is now with our production department. 

Kind regards, 

on behalf of

Dr. Vanesa Magar 

Academic Editor

PLOS ONE